# Human Papillomavirus in Breast Carcinogenesis: A Passenger, a Cofactor, or a Causal Agent?

**DOI:** 10.3390/biology10080804

**Published:** 2021-08-20

**Authors:** Rancés Blanco, Diego Carrillo-Beltrán, Juan P. Muñoz, Alejandro H. Corvalán, Gloria M. Calaf, Francisco Aguayo

**Affiliations:** 1Programa de Virología, Instituto de Ciencias Biomédicas (ICBM), Facultad de Medicina, Universidad de Chile, Santiago 8380000, Chile; rancesblanco1976@gmail.com (R.B.); diegocb17@hotmail.com (D.C.-B.); 2Instituto de Alta Investigación, Universidad de Tarapacá, Arica 1000000, Chile; jp_182_mb@hotmail.com (J.P.M.); gmc24@cumc.columbia.edu (G.M.C.); 3Advanced Center for Chronic Diseases (ACCDiS), Pontificia Universidad Católica de Chile, Santiago 8320000, Chile; acorvalan@uc.cl; 4Center for Radiological Research, Columbia University Medical Center, New York, NY 10032, USA; 5Universidad de Tarapacá, Arica 1000000, Chile

**Keywords:** breast, human papillomavirus, cancer

## Abstract

**Simple Summary:**

Breast cancer (BC) is the most frequent tumor in women worldwide. A minority of BC patients have a family history of the disease, suggesting the importance of environmental and lifestyle factors. Human papillomavirus (HPV) infection has been detected in a subset of tumors, suggesting a potential role in BC. In this review, we summarized relevant information in respect to this topic and we propose a model of HPV-mediated breast carcinogenesis. Evidence suggests that breast tissue is accessible to HPV, which may be a causal agent of BC in a subset of cases.

**Abstract:**

Breast cancer (BC) is the most commonly diagnosed malignancy in women worldwide as well as the leading cause of cancer-related death in this gender. Studies have identified that human papillomavirus (HPV) is a potential risk factor for BC development. While vaccines that protect against oncogenic HPVs infection have been commercially available, global disparities persist due to their high cost. Interestingly, numerous authors have detected an increased high risk (HR)-HPV infection in BC specimens when compared with non-tumor tissues. Therefore, it was suggested that HR-HPV infection could play a role in breast carcinogenesis in a subset of cases. Additional epidemiological and experimental evidence is still needed regarding the role of HR-HPV infection in the development and progression of BC.

## 1. Breast Cancer as a Global Health Concern

### 1.1. Breast Cancer Epidemiology

Breast cancer (BC) is the most commonly diagnosed malignancy in women worldwide as well as the leading cause of cancer-related death in this gender. It was estimated that 2,261,419 new cases of BC and 684,996 deaths worldwide were attributable to this malignant tumor in 2020 [1]. Developed parts of the world such as Australia/New Zealand, Northern and Western Europe, and North America display high incidence and low mortality rates of BC when compared with developing areas such as Melanesia, Micronesia/Polynesia, Northern and Western Africa, and the Caribbean [2]. In fact, the age-standardized incidence and mortality rates for BC in women were different when data from high-income economies and low/middle income economies (55.9 vs. 29.7 per 100,000 and 12.8 vs. 15.0 per 100,000, respectively) were compared [1]. Despite recent advances in the treatment of primary tumors, 20–40% of BC patients eventually develop recurrences depending on their stage at diagnosis and the molecular classification of the tumor [3,4].

### 1.2. Breast Cancer Classification

Approximately 90–95% of BCs are adenocarcinomas, which arise from the cells lining the milk ducts or lobular (milk-producing) glands. Of these, 75–80% are invasive ductal carcinomas (IDC) and 5–10% are invasive lobular carcinomas (ILC). Special types of invasive BC include mixed (ductal and lobular), inflammatory, metaplastic, papillary, tubular, adenoid cystic, cribriform, medullary, and mucinous carcinomas [5,6]. Based on the immunohistochemical expression of estrogen (ER), progesterone (PgR), and the epidermal growth factor receptor type-2 (HER2), BC is divided into four molecular subtypes with distinct prognostic and therapeutic approaches. The estimation of cell proliferation rate by the means of Ki-67 expression is also used for BC molecular classification. Ki-67 is a nuclear protein present in all active phases of cell cycle (late G1, S and G2/M), but absent in G0 [7]. The subtypes are as follows: Luminal A (ER+ and/or PgR+, HER2-, Ki-67 < 14%), Luminal B (ER+ and/or PgR+, HER2-, Ki-67 high or ER+ and/or PgR+, HER2+, Ki-67 any), HER2-enriched (ER-, PgR-, HER2+) and triple negative breast carcinoma (TNBC) (ER-, PgR-, HER2-) [8]. Interestingly, it was reported that approximately 25% of BCs change the molecular subtype in the same patients during tumor progression. ER and PgR usually decrease, while Ki-67 increases in recurrences compared to primary tumors, representing a change to a more aggressive BC subtype (e.g., change from Luminal A to TNBC) [9,10]. Luminal A tumors are typically the most frequent molecular subtype of BC (~50–60%), showing a better prognosis [11]. Conversely, TNBC (~15% of BC) occurs in younger women and exhibits a high histological grade, increased rates of distant metastasis, and poor prognosis. TNBC lacks the benefits of specific therapies such as hormonal therapy or anti-HER2 strategies (e.g., trastuzumab) and displays a high recurrence rate after chemotherapy [12,13]. For these reasons, research efforts are still focused on developing a better understanding of BC biology and the identification of both new molecular targets for its treatment and drug resistance biomarkers [14,15].

### 1.3. Breast Cancer Risk Factors

Age, family history, use of exogenous estrogen, and lifestyle (e.g., alcohol consumption, tobacco use) are important risk factors of BC [16]. A cohort study in the United Kingdom (UK) demonstrated that women with one first-degree relative with BC have a higher risk of developing this disease than women without any affected relatives. Furthermore, the risk escalates even higher in women with two or more first-degree relatives with BC [17]. The inherited susceptibility to BC is partially attributed to the mutations of BC-related genes such as BRCA1 and BRCA2 [18]. A study has identified more than 20 common genetic variants, which individually alter BC risk and it was reported that other gene mutations also increase the risk for BC such as TP53, PTEN, STK11, ATM, CHEK2, BRIP1, RAD51C, RAD51D, BARD1, and PALB2 [19]. In addition, approximately 90 more frequent variants with low penetrance have also been identified in genome-wide association studies (GWAS) [20]. Several studies have investigated whether the relative risks associated with common genetic BC susceptibility loci are modified by environmental risk factors [18,21,22,23,24]. The identification of the gene-environment and its interaction may help to understand BC etiology and its biological pathways. Using pooled data from 24 studies from the BC Association Consortium, it was investigated whether the impact of these genetic variants is influenced by environmental factors such as parity, body mass index, height, oral contraceptive usage, menopausal hormone therapy use, alcohol intake, cigarette smoking, and physical activity, all of which are known to influence the risk of developing BC [18].

Some viral infections have been proposed as risk factors for sporadic BC. For instance, the mouse mammary tumor virus (MMTV) is almost absent in hereditary BC compared with 30% of sporadic BC [25]. The low prevalence of MMTV could be involved in exacerbating BC, though various studies have reported contradicting results [26,27]. Epstein-Barr virus (EBV) is a persistent virus associated with the development of Burkitt lymphoma, post-transplant lymphoma, nasopharyngeal carcinoma, Hodgkin’s disease, and gastric cancer [28,29,30,31]. Although EBV presence has been reported in BC at different frequencies in different countries, a causal role of EBV in BC is controversial [32,33]. In a study from India, EBV-encoded RNA in situ hybridization (EBER-ISH) was positive in 30.1% of BC cases [34], while a study from China detected EBV in 60% of the studied BC cases using multiplex polymerase chain reaction (PCR) [35]. European studies demonstrated EBV positivity in 25.8% (Portugal) and 33.2% (France) of breast tumors using real-time PCR technique [36]. A lower prevalence of EBV (8%) was detected in BC from Iran using real-time PCR [37]. However, other studies failed to detect EBV in investigated BC cases [38,39,40]. In addition, it has been suggested that high-risk human papillomavirus (HR-HPV) infection is a risk factor for developing BC. HR-HPV types have the potential to induce the malignant transformation of epithelial cells and cancer development (e.g., cervical, oropharyngeal, and anogenital tumors). The association between HR-HPV infection and BC development will be addressed in the subsequent sections.

## 2. Human Papillomavirus (HPV)

### 2.1. Genome Organization, Structure, and Replication Cycle

HPV is a non-enveloped virus which belongs to the Papillomaviridae family and exclusively infects epithelial cells. The HPV genome comprises 8 kb double-stranded circular DNA containing 8 protein-coding genes. Its genome is functionally divided into three main regions: early (E), late (L), and the long control region (LCR). The E region contains open reading frames (ORFs) which encode the E1 to E7 non-structural proteins. These proteins are implicated in both viral replication and the transformation of host cells. Additionally, the L region encodes for two structural proteins, the major capsid protein (L1) and the minor capsid protein (L2) (reviewed in [41]) (Figure 1). Based on the L1 sequence, more than 200 HPV types have been identified. These types are divided according to the oncogenic potential in HR-HPV (e.g., 16, 18, 31, 33, 34, 35, 39, 45, 51, 52, 56, 58, 59, 66, 68, and 70) and low-risk (LR)-HPV types (e.g., 6, 11, 42, 43, and 44) [42,43]. The LCR is located between L1 and E6 genes and is approximately 1000 bp in size. This region can be divided in three segments: (1) the 5’ segment of LCR which contains a nuclear matrix attachment region, transcription termination signals and polyadenylation sites; (2) the central or enhancer segment which is involved in the regulation of viral gene expression; and (3) the 3’ segment that contains the sequence where DNA viral replication is initiated (origin of viral replication) (Reviewed in: [44]). Generally, HPVs infect the basal cells of the stratified epithelium through microlesions which occur during sexual intercourse. Heparan sulfate proteoglycans (HSPG), such as glypicans and syndecans, are required for HPV entry [45]. In addition, integrin complex α6β4 and tetraspanins are candidates for the uptake receptor complex [46,47]. The virus is internalized by a macropinocytosis-like mechanism [48] and transported into the nucleus. In undifferentiated basal cells, HPV is maintained at a low-copy number (approximately 50–100 copies per cell) [49]. The productive phase of the HPV life cycle occurs in differentiated epithelial cells (suprabasal layers), in which DNA replication activity is suppressed. In these cells, HPV E6 and E7 oncoproteins induce the inactivation of p53 and retinoblastoma (Rb) tumor suppressors, respectively, enabling the cells to retain the capacity for DNA replication. This fact results in the amplification of viral genomes copies per cell [50]. Afterward, the expression of late genes is induced, viral genome encapsidation occurs, and progeny virions are released from the cornified keratinocytes.

### 2.2. HPV Oncoproteins

The biological functions of each HPV oncoprotein are summarized in Table 1. E1 and E2 are related to viral DNA replication and the regulation of early transcription [51,52]. The E4 protein shows multiple roles during HPV replication, being likely related to efficient viral release (reviewed in [53]). The interaction of E5, E6, and E7 viral proteins with host proteins induces cell transformation and immortalization. E5 has been found to be mainly related to the expression and activation of the epidermal growth factor receptor (EGFR) [54,55]. E5 protein interacts with the 16K subunit of the vacuolar (H+)-ATPases (V-ATPases), disrupting the normal v-ATPase-dependent endosomal acidification process, which results in reduced EGFR degradation [56,57]. Additionally, it was demonstrated that HPV18 E5 protein enhances EGFR expression in primary human keratinocytes [55]. Moreover, HPV16 E5 disrupts the EGFR ubiquitination in human foreskin keratinocytes, which delays the EGFR degradation upon activation and increases the amount of cell surface receptors [54]. Indeed, the E5 knockdown reduced both EGFR and pEGFR in E5/E6/E7 co-expressing cells [58], which support the role of E5 protein in promoting EGFR signaling pathway deregulation. Alternately, HPV16 E5 downregulates the major histocompatibility complex class I (MHC-I), promoting immune evasion [59]. E6 and E7 are considered the major HPV oncoproteins (reviewed in [41]). E6 promotes the inactivation of p53 through E6-associated protein (E6-AP)-mediated ubiquitination and its subsequent proteasome degradation, thus protecting infected cells from apoptosis [60] (Figure 2, left). E6 also disrupts the apoptotic program targeting the pro-apoptotic Bak for degradation [61] and Fas-associated protein with death domain (FADD) and caspase-8 [62]. Importantly, E6 can activate both the Mitogen-activated protein kinase (MAPK) and the mechanistic target of rapamycin complex 1 (mTORC1) pathways [63,64]. Furthermore, HPV16 E6 reduces interleukin (IL)-1β expression by proteasomal degradation of pro-IL-1β, which is necessary for the activation of Th1 CD4+ T cells [65]. MHC-I expression is also reduced by HPV38 E6, through the signal transducer and activator of transcription 1 (STAT1) downregulation [66]. E7 targets Rb protein for ubiquitination, leading to E2F transcription factor release, forcing cell entry to S-phase [67,68] (Figure 2, right). E7 can bind the stimulator of interferon genes (STING) decreasing the production of interferons (INFs) [69]. Additionally, HPV16 E7 increases EGFR expression and upregulates the protein kinase B (PKB or AKT) activity [70,71]. Both E6 and E7 proteins are also able to interact with c-Myc oncogene. This interaction induces the activation of the human telomerase reverse transcriptase (hTERT) promoter and hTERT expression, contributing to the evasion of cellular senescence [72,73]. Moreover, both E6 and E7 promote cell migration and invasiveness by upregulating N-cadherin, Fibronectin and Vimentin, which are involved in the epithelial-to-mesenchymal transition (EMT) [74]. The capacity of E6 and E7 to inhibit the production of antiviral IFNs and the interleukin-1 beta (IL-1β) secretion by macrophages facilitates infected cells in evading the host immune system [75].

## 3. HPV Infection and Breast Cancer

### 3.1. Routes for HPV Infection in Breast

The possible route of HPV infection to the breast is yet to be fully known. Lawson et al. found the same HR-HPV genotype in both BC specimens and squamous intraepithelial lesions of the cervix (SILs) counterparts from the same patients [76]. Additionally, the presence of HPV infection was reported in BC samples from women with HPV-associated SILs [77,78]. Taken together, these findings allowed the authors to suggest that HPV could reach the breast tissue through circulation (blood or lymphatic systems) in patients who displayed HPV-positive cervical cancer [79]. Bodaghi S et al. (2005) and others suggested that HPV could be transmitted through the bloodstream because HPV DNA was found in peripheral blood mononuclear cells (PBMC) [80]. However, all these findings failed to demonstrate that infectious HPV was present in the blood or that PBMC cells are susceptible to or permissive for HPV infection. The evidence shows that basal cells in a stratified epithelium are susceptible to HPV infection, with only cells from the upper highly differentiated epithelium strata, being permissive. Whether HPV virions are transported to the bloodstream from infected sites to reach the breast or any other anatomical region, has not yet been demonstrated and remains a matter of speculation. Therefore, as HPV replication occurs in the stratified epithelium as a mechanism to evade immune recognition, a route through the bloodstream seems difficult to sustain from a virological point of view. It has also been suggested that free HPV DNA in blood could reach the breast, which can incorporate this viral DNA through transference [78,81]. Interestingly, De Carolis et al. reported that HPV DNA is found in the serum-derived extracellular vesicles (EV) and may be transferred to TNBC. The authors suggested the possibility that HPV DNA from EV can confer an increased aggressive feature to the BC [82,83]. However, further studies are needed to clarify whether HPV DNA is transported from HPV infected sites, through EV, to this anatomical site. Another theory proposes that HPV could reach the breast tissue through nipple or areola microlesions of breast skin that may occur during oral-breast or genital-breast sexual intercourse [79]. Therefore, considering that HPV infection requires direct contact with cutaneous or mucosal epithelium, this transmission route to the breast appears to be the most feasible alternative for HPV entry to breast cells. We cannot deny the possibility that both transmission routes (sexual and EV) and others may be involved. A general scheme showing breast cancer transmission routes is shown in Figure 3.

### 3.2. Epidemiology of HPV Infection in Breast Cancer

The prevalence of HPV in BC ranges 1.6–86.2% worldwide [84,85], although some authors reported a lack of HPV infection in these tumors [86,87,88]. HPV frequency in breast cancer from different continents did not show significant differences [79]. Note that comparisons among studies are difficult to establish because differences in specimens (fresh frozen tissue, paraffin embedded tissue, etc.) and methodologies showing variable sensitivity and specificity (e.g., PCR with different primers) for HPV detection. In addition, we cannot deny the possibility of contamination with previous PCR products that may have affected some results during both preanalytical and analytical phases. Despite the above, some geographic HPV distribution has been reported. For instance, HPV positivity was evidenced in 32.42% and 12.91% of BC patients from Asia and Europe, respectively [89]. In addition, 42.9% of HPV was detected in BC individuals from North America and Australia, and 15.1% in Central America and South America [90]. A study based on The Cancer Genome Atlas (TCGA) database and analyzed by Next-Generation Sequencing (NGS) with data from Australian BC specimens, reported the presence of HR-HPVs in 2.3% of BCs. Importantly, the authors found a correlation between HR-HPV presence in benign breast specimens and HR-HPV positive BCs in the same patients. In addition, HR-HPV in BC was detected to be biologically active [91]. A meta-analysis conducted by Bae and Kim (2016) found a 4.02-fold (95 % CI: 2.42–6.68) increased risk for BC development in HPV-positive individuals [92]. Similarly, Choi et al. (2016) reported an association between BC and HPV infection (OR = 5.43, 95% CI: 3.24–9.12) [93]. Furthermore, a similar study published by Ren et al. (2019) which included 3607 BC cases and 1728 controls showed a statistically significant association of BC development and HPV infection (OR = 6.22, 95% CI: 4.25–9.12) [94]. Interestingly, HPV infection was demonstrated in BC samples but not in the normal tissues [95]. Conversely, HPV infection was also reported in normal breast samples, though it was significantly decreased compared to BC [96,97]. Additionally, an increased frequency of HPV infection was found in BC when compared with benign breast lesions such as fibroadenomas, fibrocystic changes, mastitis [98], intraductal papilloma [99,100] and breast adenosis [101]. HPV infection was also significantly increased from adjacent normal breast (9.5%) and fibroadenomas (30%) to BC (64.8%) samples [102]. The presence of koilocytes was evidenced in breast cancer cells, which matched with HPV-positive cells by in situ PCR [103]. To date, mostly HR-HPV types (HPV16, 18, 31, 33, 35, 45, and 52) have been detected in breast carcinoma samples (reviewed in [79]). However, infection with low-risk subtypes (HPV6 and 11) has also been reported [97,104]. HPV16 is by far the most frequent genotype found in BC patients, followed by HPV18 and 33 (reviewed in [79]). In fact, genotypes HPV16 and 18 were detected in 87.5% and 12.5% of HPV-positive BCs, respectively [98]. HPV16 was also reported in 77.37% of HPV-positive BCs followed by HPV33 (13.64%) and HPV31 (9.09%) [105]. Nevertheless, some authors have found HPV33 [106], HPV39 [107] or HPV51 [108] to be the most prevalent HPV genotypes in BC tissues. A preferential presence of HPV DNA was reported in high grade BC (II/III) [109]. Moreover, HPV16 and 58 were mostly detected in grade II BC, while HPV18 infection was increased in grade III tumors [110]. According to the molecular classification, HPV infection was evidenced in all subtypes of BC (Luminal A, Luminal B, HER2-enriched, and TNBC) (reviewed in [79]). However, an increased presence of HPV was reported in TNBC, and HER2-enriched tumors, compared to Luminal A and B types [82,111]. TNBC and HER2-enriched tumors are associated with more aggressive biological behavior of BC. HPV infection was also found to be increased in hormone receptor-positive BCs compared with HER2-positive tumors [99]. Furthermore, the presence of HPV was particularly detected in HER2-negative Luminal B tumors, although no statistically significant difference was reached when compared to the rest of the molecular subtypes, including Luminal B/HER2+ phenotype [104]. Interestingly, in Luminal A and Luminal B tumors, the presence of HPV DNA was related to the extent of lymph node invasion and increased proliferation rate, respectively [82]. Altogether, these facts suggest an association between HPV infection and more aggressive forms of BC disease. However, no association with any particular molecular subtype was reported [105].

### 3.3. Role of HPV Infection in Breast Carcinogenesis

As koilocytes have been detected in normal breast skin and BC tissue, it was suggested that HPV virions can reach the breast [112]. HR-HPV genomes present in BC have been mostly detected in an integrated physical status (86–100%) [97,102,113,114]. For instance, Islam et al. reported a statistically significant HPV16 integration in BC samples when compared to the episomal form (87.5% vs. 4.2%; *p* = 0.01), while both episomal and integrated forms (mixed) were evident in 8.3% of BCs from Indian patients [102]. It is well established that HR-HPV integration is an important step for epithelial cell carcinogenesis [115], although this is not a requisite [116]. Other mechanisms of HPV-mediated oncogenesis have been described in tumors harboring episomal forms of HPV [117,118]. Regarding viral load, it has been reported that this is extremely low, including levels less than 1 viral genome per cell. Considering that at least 1 copy/cell is required for clonal viral presence in each tumor cell (as occurs in SiHa cervical cancer cell line, which harbors 1 copy of integrated HPV16), it seems difficult to sustain a scenario where all the BC cells harbor integrated HPV [114,119]. Thus, the low viral load in BCs suggests the possibility that the role of HPV in these tumors is only indirect. This notion is supported by the fact that BC risk is not enhanced in immunosuppressed patients or organ transplant recipients [119]. Another possibility is that the restricted HPV positive population can cause molecular alterations for BC initiation and subsequently, HPV presence is no longer required, compatible with a “hit and run” mechanism [120]. However, low copies of the HPV genome per cell (<1 copy/cell) seem to be sufficient to induce a premalignant phenotype [121,122]. It was reported that HPV-positive BC shows both increased proliferation rates and histological grade when compared to HPV-negative tumors [123]. Moreover, the expression of the tumor suppressors BRCA1 and BRCA2 was decreased in patients with HPV-positive BC. These authors also found the expression of some proinflammatory cytokines such as IL-1, TGF-β, and TNF-α, related to HPV infection, which are also involved in tumor progression [97]. HPV infection was also correlated with p-Stat3 and IL-17 expression in BC patients [101]. Additionally, the expression of HPV E6 and E7 oncoproteins has been found in BC [91,120]. A low expression of HPV E6/E7 proteins has also been detected in BC, most likely due to the hypermethylated state of the HPV16 p97 early promoter [102]. Nevertheless, the expression of E6 and E7 was negatively associated with p53 and Rb, respectively. In fact, the expression levels of p53 and Rb were decreased in HPV-positive BC samples compared to HPV-negative tumors [97]. Similarly, Wang et al. found a significantly higher Bcl-2 and lower p53 expression in HPV-positive BC compared to HPV-negative tumors, although the expression of p21, Rb, and survivin was not associated with HPV presence [124]. The expression of HPV E6 was associated with the overexpression of DNA-binding protein inhibitor (Id-1) in invasive BC [106]. BC cells expressing HPV16 E6/E7 display an increased invasiveness and in vivo metastatic potential when compared to wild type cells [125]. HPV16 E6/E7 cooperates with HER2 to increase the EGF-independent colony formation in normal human mammary epithelial cells [126]. Moreover, HPV16 E6/E7 also cooperates with HER2 to promote cell migration and colony formation; it also activates beta-catenin, a regulator of cell adhesion [127]. It was demonstrated that HPV18 upregulates the expression of the DNA cytosine deaminase APOBEC3B protein, increasing γH2AX focus formation and DNA double strand breaks [128]. APOBEC3B acts as an enzymatic source of DNA damage and mutation in BC [129]. HPV16 E6 and E7 interact with BRCA1, disrupting the suppression of ER-α activity in BC cells [130]. In addition, HPV16 E6 promotes BC cells proliferation and anchorage-independent growth by upregulation of Cyclooxygenase-2 (COX-2) expression. COX-2 overexpression mediated by HPV16 E6 has also been associated with the activation of NF-κB signaling pathway [131]. Conversely, absence of HPV E6 and E7 expression was also reported in HPV positive BC samples [132,133]. Recently, it was found that HPV oncoproteins cooperates with EBV for breast carcinogenesis. The authors demonstrated that E6/E7 cooperate with EBV LMP1, increasing invasion and EMT in breast cancer cells [134]. However, LMP1 is absent [135] or weakly detected in EBV positive BCs [136], thus additional studies are warranted to explore the possibility of a E6/E7/LMP1 cooperation. A potential cooperation between HR-HPV E6/E7 oncoproteins and EBV BamHI-A rightward frame 1 (BARF1) protein presents itself as an interesting possibility to be evaluated in the future because BARF1 is considered an exclusively epithelial EBV oncogene [137,138].

### 3.4. HPV and Its Association with Estrogen Receptor Signaling

Exposure to estrogens has been widely shown to increase the risk of cervical, endometrial, and BC [139,140,141]. However, the molecular basis behind this phenomenon remains unclear. 17β-estradiol, the most common type of circulating estrogen, is a key steroid hormone for human physiology that exhibits a plethora of biological and physiological functions through its interaction with the estrogen receptor (ER) [142]. ERs are ligand-inducible transcription factors that belong to the nuclear steroid hormone receptor superfamily and are overexpressed in 60–70% of human BCs [143]. Two ER isoforms have been described, ERα and ERß. Although they are encoded by two distinct genes, which both exhibit similar functional and structural organization. The transmission of the estrogen signaling includes activation of estrogen receptors and signal transduction, which can be mediated by genomic and non-genomic pathways. The genomic pathway is the most characterized type of ER signaling and is initiated by their ligand binding, which induces a conformational change, leading to translocation of ER from the cytoplasm to the nucleus and binding to specific DNA sequences. These sequences are called Estrogen Response Elements (EREs) and are located in or near the promoters of target genes [144]. Consequently, the variety of transcriptional regulation mechanisms in different cells by ERs and their interactions with local transcription factors lead to the continuous altering of gene expression [144]. In cervical cancer, ER-mediated signaling has been shown to promote cell transformation in combination with HPV oncogenes [145,146]. Based on experiments carried out in the HPV transgenic mouse model, estrogens treatments induce overexpression of E6 and E7 and an increased severity of cervical dysplasia [147]. Therefore, a synergy between estrogen signaling and HPV-mediated oncogenesis has been widely validated [148,149]. Considering the evidence of integrated forms of the HR-HPV genome in mammary cells, different oncogenesis mechanisms enabled by a cooperation between estrogens and HPV could be hypothesized in breast cells. For instance, ERα can interact with the transcriptional factors c-Fos and c-Jun at their binding regions, which are key regulators in the LCR of HPV [150]. Therefore, the presence of HPV genomes in mammary cells, which are highly exposed and sensitive to the supply of estrogens, could produce a favorable environment for the promotion of viral gene expression, leading to aberrant overexpression of E6 and E7. Conversely, HPV could have a potential effect on the ER signaling. In fact, Wu et. al. have shown that E2 protein cooperates with nuclear receptor co-activators to increase the ERE-dependent transcriptional activity of ERα [151]. Likewise, Wang et al. demonstrated that HPV-18 E6 and E7 oncoproteins can directly interact with ERα enhancing the ERE-dependent activity [152]. Therefore, the strong estrogen signaling resulting from ER overexpression in most cases of BC, could influence the HPV gene expression in those HPV positive breast cells, favoring the initiation as well as the progression of BC. Thus, a hypothetical model of HPV-mediated breast carcinogenesis which involves the ER is proposed in Figure 4.

## 4. Conclusions and Remarks

Koilocytes presence and molecular approaches confirm that HPV virions can reach the breast, most likely through epithelium or mucosal contact during sexual intercourse. This transmission route requires microlesions in the areola for HPV entry. Importantly, HPV components such as nucleic acids can reach the breast from infected sites through the bloodstream by extracellular vesicles. The extremely low viral load detected in BC specimens indicates that HPV genomes are not clonally distributed in the tumor, although HPV is frequently integrated into the host genome. Taken together, epidemiological and experimental evidence suggests that HR-HPV may be involved in BC development in a proportion of cases. On the other hand, it is well known that HR-HPV is a necessary condition for the development of cervical cancer, though the infection itself is not sufficient to establish cell transformation, suggesting that other factors are potentially involved in HPV-driven carcinogenesis. Thus, interaction models between HPV and additional host (e.g., ER) or environmental factors (e.g., pesticides, tobacco smoking) offer a plausible way to explain the frequency of HR-HPV infection in BC. The identification of HR-HPV as a causal agent in a subset of BC patients with increased risk of disease progression will lead to prevention or therapeutic alternatives. Additional epidemiological and experimental approaches are warranted to dissect the specific molecular alterations involved in HPV-driven BC.

## Figures and Tables

**Figure 1 biology-10-00804-f001:**
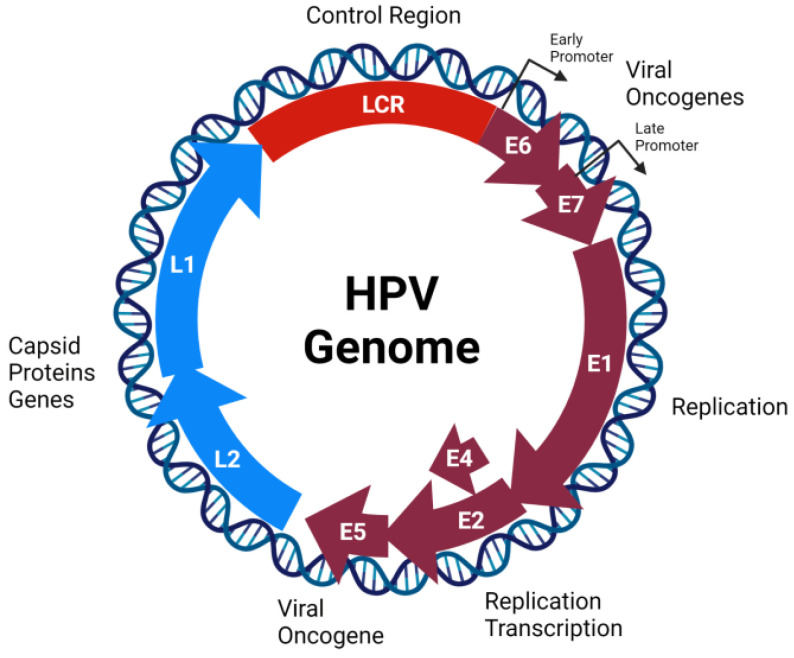
Human papillomavirus (HPV) genome organization. The HPV contains a circular double-stranded DNA genome with approximately 8000 base pairs (bp), organized into three regions. The E or early region (purple) contains the E1, E2, E4, E5, E6, and E7 open reading frames (ORFs). The L or late region (blue) contains the L1 and L2 ORFs. Finally, the Long Control Region (LCR) (red) contains regulatory elements.

**Figure 2 biology-10-00804-f002:**
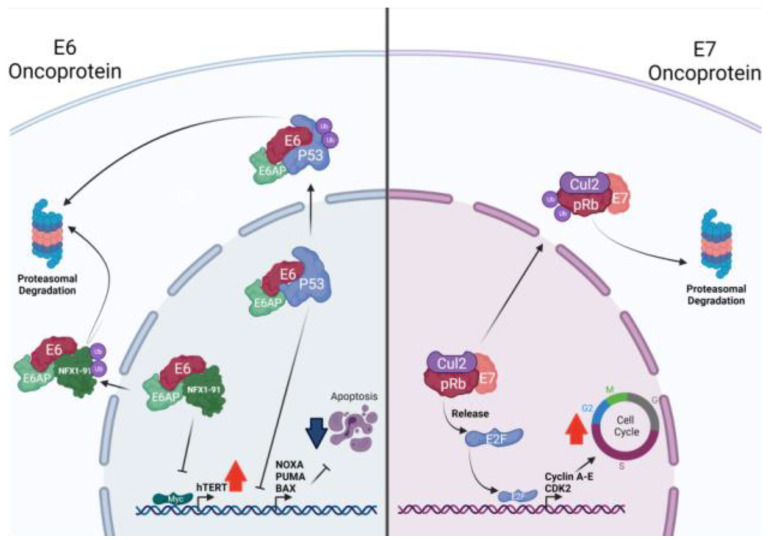
HR-HPV E6 oncoprotein induces p53 degradation, thus impairing the ability of the cell to promote apoptosis. HR-HPV E7 oncoprotein induces the Rb degradation, promoting S phase entry. The E6 protein blocks the binding of the transcription factor p53 to the promoters of pro-apoptotic genes through its binding with the ubiquitin ligase E6-AP, thus inducing its proteasomal degradation. Additionally, the E6 protein promotes transcriptional activation of the promoter of the catalytic subunit of telomerase enzyme (hTERT) in a myc-dependent manner. This mechanism is induced by the degradation of NFX1-91 mediated by the binding of E6 to E6-AP. The E7 oncoprotein induces the degradation of the retinoblastoma protein (pRb) by binding to the Cullin 2 protein (Cul2), which is part of the E3 ubiquitin ligase complex. Thus, the release of E2F occurs along with the subsequent binding as a transcription factor to the genes that regulate the cell cycle.

**Figure 3 biology-10-00804-f003:**
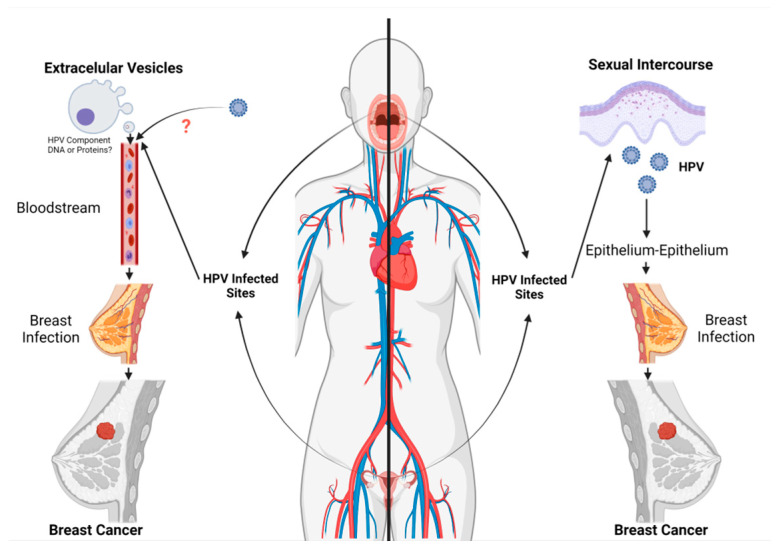
Possible HPV transmission routes. Two routes of HPV infection of the mammary epithelium are plausible. First, direct contact between an infected epithelium and the mammary epithelium. In this way, the virus can enter through microlesions that would allow infection and the expression of viral oncoproteins, which in collaboration with other factors can account for a breast tumor. On the other hand, extracellular vesicles (EV), which can be released to the bloodstream by HPV-infected cells, can transfer viral biomolecules to the breast. These biomolecules (EV cargo: proteins, nucleic acids, microRNAs) may go on to collaborate with other factors in tumor development.

**Figure 4 biology-10-00804-f004:**
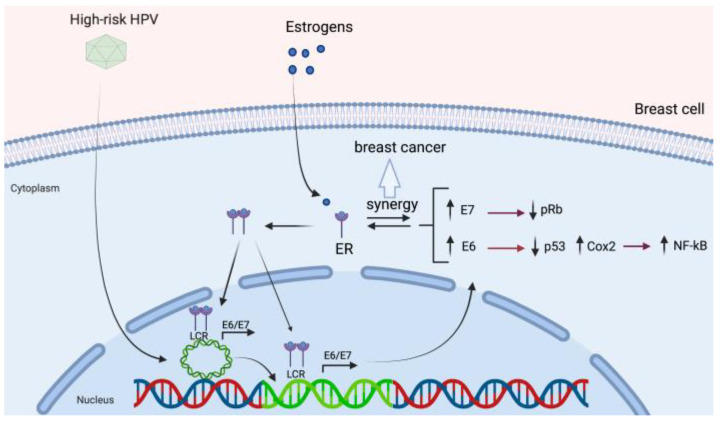
A suggested model of HPV-mediated breast carcinogenesis. Estrogens promote estrogen receptor (ER) activation and nuclear translocation. ER binding to cognate sites into the HR-HPV LCR promotes early promoter activation and E6/E7 transcript overexpression (in both episomal and integrated forms of HPV). The expressed E6 and E7 oncoproteins induce pRb and p53 downregulation, Cox-2 upregulation and NF-κB activation. Additionally, a synergism between ER and HR-HPV E6/E7 has been previously proposed.

**Table 1 biology-10-00804-t001:** Biological functions of HPV oncoproteins.

HPV Protein	Function	References
E1	Initiates viral genome replication.	[51]
E2	Induces viral DNA replication and transcription.Modulates viral gene expression.	[52]
E3	Not known.	[53]
E4	Facilitates the encapsidation and maturation of viral particles.	[53]
E5	Disrupts the v-ATPase-dependent endosomal acidification process, reducing EGFR degradation.	[56,57]
	Increases both EGFR and pEGFR expression on cell surface.	[54,55,58]
	Downregulates the MHC-I, disrupting the host immune response.	[59]
E6	Protects cells from apoptosis inducing p53, Bak, FADD, and caspase-8 degradation.	[60,61,62]
	Activates MAPK and mTORC1 signaling.	[63,64]
	Disrupts the activation of Th1 CD4+ T cells by means of pro-IL-1ꞵ degradation.	[65]
	Downregulates MHC-I expression, targeting STAT1.	[66]
E7	Induces cell cycle progression by means of Rb degradation.	[67,68]
	Decreases INFs production targeting STING.	[69]
	Activates AKT activity and EGFR expression.	[70]
E6 and E7	Interact with c-myc, inducing hTERT activation and cell immortalization.	[72,73]
	Induce EMT, upregulating N-cadherin, Fibronectin, and Vimentin.	[74]
	Inhibit INFs and IL-1β production, contributing to immune evasion.	[75]

EGFR, epidermal growth factor receptor; MHC-I, major histocompatibility complex class I; FADD, Fas-associated protein with death domain; MAPK, mitogen-activated protein kinase; mTORC1, rapamycin complex 1; IL, interleukin; STAT1, signal transducer and activator of transcription 1; STING, stimulator of interferon genes; INF, interferon; AKT, protein kinase B; hTERT, human telomerase reverse transcriptase; EMT, epithelial-to-mesenchymal transition.

## Data Availability

Not applicable.

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
