# Peer review of "Human Papillomavirus in Breast Carcinogenesis: A Passenger, a Cofactor, or a Causal Agent?"

_biology, 2021, doi:10.3390/biology10080804_

Round 1
Reviewer 1 Report
The authors addressed an important question of HPV-induced breast carcinogenesis. They tried to include some important information in the review but some areas need a major improvements. Please address the following questions in the manuscript:
- In introduction is the number of death worldwide or specific area? Mention it.
- Explain what is high-risk-HPV infection in the introduction?
- In section 1.1, is the difference between low income and high income country related to HPV infection? Please explain
- In section 1.2, what does Ki67 means and what is its role in the classification of breast cancers, please give more details
- Please insert figure legends for Fig 1, E3 protein is missing from fig 1, include and mention its role too.
- Grammatical corrections required throughout the manuscript
- In Fig 2, please insert full abbreviation in the legend, please explain the figure in the figure legend.
- Make a table listing the function of all the HPV viral proteins, summarizing 2.2
- Figure 3: Explain the figure in figure legend
- Section 3.1 is very confusing. Please make it understandable to the readers. The authors can please incorporate the HPV routes in a table in case of breast cancer and in other HPV-induced diseases
- Please correct the numbers in Line 226. It will be 3,607 and 1,728
- Explain why HPV affects breast tissue not the normal tissue
- Please make a figure illustrating breast carcinogenesis by HPV
- Please mention the significance and future perspective of this research in conclusion.
Author Response
The authors addressed an important question of HPV-induced breast carcinogenesis. They tried to include some important information in the review, but some areas need a major improvement. Please address the following questions in the manuscript:
Reviewer. In introduction is the number of deaths worldwide or specific area? Mention it.
Answer. This is worldwide (Line 34)
Reviewer. Explain what is high-risk-HPV infection in the introduction?
Answer. This was explained (Lines 105-108).
Reviewer. In section 1.1, is the difference between low income and high income country related to HPV infection? Please explain.
Answer. According to our knowledge, HPV frequency in breast cancer from different continents did not show significant differences. In addition, some HPV genotypes distributions were found, though differences in methodologies of HPV detection make difficult to establish comparisons. These sentences were added in the manuscript (lines 245-251)
Reviewer. In section 1.2, what does Ki67 means and what is its role in the classification of breast cancers, please give more details.
Answer. Ki67 is a well-known proliferation marker. Additional sentences were added in lines 53-55.
Reviewer. Please insert figure legends for Fig 1, E3 protein is missing from fig 1, include and mention its role too.
Answer. This was done. E3 protein during HPV infection has not been detected.
Reviewer. Grammatical corrections required throughout the manuscript
Answer. The manuscript was extensively checked and reviewed by a native English speaker.
Reviewer. In Fig 2, please insert full abbreviation in the legend, please explain the figure in the figure legend.
Answer. This was done.
Reviewer. Make a table listing the function of all the HPV viral proteins, summarizing 2.2
Answer. This was done (Table 1).
Reviewer. Figure 3: Explain the figure in figure legend
Answer. This was done.
Reviewer. Section 3.1 is very confusing. Please make it understandable to the readers. The authors can please incorporate the HPV routes in a table in case of breast cancer and in other HPV-induced diseases
Answer. This section was extensively edited for a better understanding, Descriptions were made refusing some suggested transmission routes, and we consider that only a transmission route through sexual intercourse (epithelial contact) and possibly, by extracellular vesicles are plausible This was clearly indicated in Figure 3. The legend of this figure was better explained.
Reviewer. Please correct the numbers in Line 226. It will be 3,607 and 1,728
Answer. This was done.
Reviewer. Explain why HPV affects breast tissue not the normal tissue
Answer. HPV affects normal breast epithelium which is susceptible and permissive for HPV infection (demonstrated by koilocytes). However, HPV-mediated breast cancer requires additional alterations that are discussed in the manuscript.
Reviewer. Please make a figure illustrating breast carcinogenesis by HPV
Answer. This was done (Figure 4). A suggested model including HPV oncoproteins and estrogen receptor interactions was included.
Reviewer. Please mention the significance and future perspective of this research in conclusion.
Answer. Sentences were added in conclusion section (lines 410-415).
Reviewer 2 Report
Blanco et al. is a thoroughly researched, well written review. This work describes what is known about the association of human papillomavirus infection and potential roles for the onset of breast cancer in infected individuals. This work is easy to understand; very well done. As such, I only have a minor comment that should be addressed prior to publication.
Minor comment:
The figures should include descriptive legends to accompany the figure.
Author Response
Reviewer. Blanco et al. is a thoroughly researched, well written review. This work describes what is known about the association of human papillomavirus infection and potential roles for the onset of breast cancer in infected individuals. This work is easy to understand; very well done. As such, I only have a minor comment that should be addressed prior to publication.
Answer. Many thanks for these comments.
Reviewer . Minor comment: The figures should include descriptive legends to accompany the figure.
Answer. The legends were significantly improved.
Reviewer 3 Report
The review is interesting, however, the evidences are not convincing regarding the participation of HPV in BC development. I recommend to include the methods used in the different papers cited and argue about the different methodologies used and the possible vias based on methodology limit, possible contamination, etc. Additionally, HPV use TFs, translation machinery and other factors that permit its reproduction that are specifically found in cells for which it has tropism. You can argue about this regard to increase your review data.
Particular observations:
In lines 105 and 106 you mention that “2.1 Genome organization, structure, and replication cycle. HPV is a non-enveloped virus which belongs to the Papillomaviridae family and exclusively infects epithelial cells.”
I would recommend to review the receptors and molecules used by HPV to enter the cell because it infects different types of cells but it reproduces preferentially in cervix, anus and penis cells.
In lines 195 and 196 you mention “Additionally, it has been suggested that free HPV DNA in blood could reach the breast, which can incorporate this viral DNA through transfection”.
Transfection is a laboratory methodology in which nucleic acids are introduced into cells by liposomes or other lipid components. I recommend to change transfection for transference.
Author Response
Reviewer. The review is interesting, however, the evidences are not convincing regarding the participation of HPV in BC development. I recommend to include the methods used in the different papers cited and argue about the different methodologies used and the possible vias based on methodology limit, possible contamination, etc. Additionally, HPV use TFs, translation machinery and other factors that permit its reproduction that are specifically found in cells for which it has tropism. You can argue about this regard to increase your review data.
Answer. The manuscript was extensively checked. A sentence with methodological considerations was added (line 244-248)
Reviewer. Particular observations: In lines 105 and 106 you mention that “2.1 Genome organization, structure, and replication cycle. HPV is a non-enveloped virus which belongs to the Papillomaviridae family and exclusively infects epithelial cells.”I would recommend to review the receptors and molecules used by HPV to enter the cell because it infects different types of cells but it reproduces preferentially in cervix, anus and penis cells.
Answer. The receptors that HPV uses for entry were mentioned in the section 2.1 (lines 128-130).
Reviewer. In lines 195 and 196 you mention “Additionally, it has been suggested that free HPV DNA in blood could reach the breast, which can incorporate this viral DNA through transfection”. Transfection is a laboratory methodology in which nucleic acids are introduced into cells by liposomes or other lipid components. I recommend to change transfection for transference.
Answer. This was changed.
Reviewer 4 Report
The present manuscript titled "Human papillomavirus in breast carcinogenesis: a passenger, a cofactor, or a causal agent," authors have discussed role of human papillomavirus (HPV) in BC development. Recently, there have been research which pointed out that HPV infection could be a potential risk factor for development of breast cancer. Authors have summarized the available studies and evidences and presented it well. It is well written and organized review article.
- It will be good if you can present breast cancer subtypes and causal factors in tabular forms.
- In the figure 2, text (labels) are not clear.
Author Response
The present manuscript titled "Human papillomavirus in breast carcinogenesis: a passenger, a cofactor, or a causal agent," authors have discussed role of human papillomavirus (HPV) in BC development. Recently, there have been research which pointed out that HPV infection could be a potential risk factor for development of breast cancer. Authors have summarized the available studies and evidences and presented it well. It is well written and organized review article.
Reviewer. It will be good if you can present breast cancer subtypes and causal factors in tabular forms.
Answer. Many thanks fos this observation. This section was improved for a better understanding (English grammar and some additions). Because and additional table and Figure were included, we considered unnecessary to include a specific table for this purpose.
Reviewer. In the figure 2, text (labels) are not clear.
Answer. This was corrected.